# Environmental mastery mediates relationships between mindsets and well-being

John B. Nezlek [1,2]*, Marzena Cypryańska[1], Joanna Gutral[1,3]

1 Center for Climate Action and Social Transformations (4CAST), Institute of Psychology, SWPS University, Warsaw, Poland, 2 Department of Psychological Sciences, College of William & Mary, Williamsburg, Virginia, United States of America, 3 Department of New Technologies and Social Applications, Faculty of Design, SWPS University, Warsaw, Poland

* jbnezl@wm.edu

## Abstract

Research on mindsets, the extent to which people believe that people can change (incrementalism) has found that incrementalism is positively related to success in various domains. One explanation for this relationship is that incrementalism is associated with a mastery orientation, which in turn is associated with success/achievement. The present study examined if the relationship between incrementalism and positive outcomes can be extended to include well-being, and if so, would a mastery orientation mediate such relationships. The present study examined if environmental mastery as conceptualized by Ryff and colleagues mediated relationships between mindsets and well-being. Participants ($n = 447$) completed measures of implicit theories of the self (incrementalism), Ryff and Keyes's multidimensional measure of well-being, meaning in life, positivity, optimism, future time perspective, and self-esteem. A series of mediational analyses found that environmental mastery mediated relationships between incremental beliefs and all measures of well-being. For most measures of well-being, the direct effects of mindset beliefs on well-being were rendered non-significant when environmental mastery was included as a mediator. The present results confirm and extend to the general domain the supposition that a mastery orientation is responsible for relationships between well-being and incremental theories of the self.

## Introduction

The present study had two goals: (1) to examine the possibility that people's beliefs that people can change, what Dweck and colleagues refer to as incrementalist mindset [e.g., 1], is positively related to well-being, (2) to examine the possibility that relationships between incrementalist beliefs and well-being are mediated by individual differences in people's mastery orientations. These possibilities were examined in a cross-sectional design in which participants completed measures of mindsets,

**Data availability statement:** All data are freely available via the Open Science framework. The site for this study includes a fully annotated SPSS file and an accompanying codebook, and a csv data file. The url is: https://osf.io/r9u3n/?view_only=9b6a7de60a004482ab7c-5038dc30de66.

**Funding:** Publication funded by the SWPS University Research Development Fund, organised by the Office of Research Strategy and Development.

**Competing interests:** The authors have declared that no competing interests exist.

well-being, and mastery orientation. Relationships between mindsets and well-being were examined via correlations, and a second set of analyses examined the extent to which mastery orientation mediated these relationships.

## Mindsets and well-being

The study of mindsets (also referred to as theories of the self or implicit theories) has focused primarily on relationships between mindsets and achievement in different domains, and much of this research has concerned achievement in the academic/educational domain. Moreover, some reviews have concluded that the extent to which someone holds incrementalist beliefs is positively related to success/achievement [e.g., 2], although there have been some concerns about the strength and consistency across situations of this relationship.

For example, Sisk et al. [3] in their meta-analysis of 273 studies concluded that: "The meta-analytic average correlation between growth mind-set and academic achievement was very weak - r =.10. … However, the overall effect is overshadowed by the high degree of heterogeneity." (p. 561). Moreover, Sisk et al. did not find any moderating variables that could account for differences in the strength of these relationships. Sisk et al. reached a similar conclusion in their second meta-analysis of 29 intervention studies (38 independent samples, 43 effect sizes): "Similarly, the second meta-analysis demonstrated only a very small overall effect of mind-set interventions on academic achievement" (p. 569). Although the meta-analysis of Burnette et al. [4] did not focus on direct relationships between mindsets and achievement, they also reported a relatively weak relationship between measures of the two constructs ($r=.095$).

Although the results of Sisk et al. and Burnette et al. [4] suggest that mindsets may not be strongly related (if at all) to academic achievement, this does not preclude the possibility that they are related to other types of positive outcomes such as well-being. Such a possibility is supported by the results of a meta-analysis of relationships between mindsets and psychological distress conducted by Burnette et al. [5]. Specifically, Burnette et al. [5] reported that growth mindsets were negatively related to psychological distress ($r=-.220$) and were positively related to the benefits people derived from treatment ($r=.137$) and to coping ($r=.207$). Although not a formal meta-analysis, Huang et al. [6] discussed a series of studies that found that growth mindsets (incrementalism) were positively related to well-being (e.g., depression and anxiety). Most of the studies Huang et al. discussed were included in Burnette et al.'s [5] meta-analysis.

Due to their relevance to the issues we discuss in this paper, we mention two studies that were not included in the meta-analysis of Burnette et al. [5] probably because they did not meet their criteria for inclusion. In a study of first year collegians over three years, Robins and Pals [7] found a negative correlation between self-esteem and the extent to which participants possessed an entity mindset. Along the same lines, in a series of studies, Howell et al. [8] found that incrementalist beliefs about well-being were positively related to both hedonic (e.g., affect balance) and eudaimonic (e.g., personal growth) well-being.



Complementing research on relationships between mindsets and well-being is the extensive research on the positive relationships between psychological flexibility (which can include mindsets) and well-being. This research has led to the proposal that psychological flexibility is a "fundamental aspect of health" [9]. Along similar lines, research on what is called coping flexibility, defined as "the ability to discontinue an ineffective coping strategy (i.e., evaluation coping) and produce and implement an alternative coping strategy (i.e., adaptive coping)" suggests that flexible coping is positively related to well-being [10].

Although psychological flexibility and coping flexibility concern a different level of analysis than incrementalist mindsets (they focus more on flexibility across situations, whereas incrementalism is a more trait-like characteristic), what these constructs have in common with incrementalism is a focus on the ability to change how one approaches challenges and tasks, which is related to adaptation, and by extension, well-being. For example, two questions on the commonly used measure of mindsets that we used [11] are: "Everyone is a certain kind of person, and there is not much that can be done to really change that" and "Everyone, no matter who they are, can significantly change their basic characteristics." These items can be thought of as measures of flexibility. Based in part on the research on the relationships between flexibility and well-being, we expected that well-being would be positively related to how incrementalist people's mindsets were.

## Defining well-being

For present purposes, we defined well-being broadly, in terms of both hedonic and eudaimonic well-being. Defining well-being in terms of these two categories has been discussed by leading motivational theorists. For example, Ryan and Deci [12] noted: "Current research on well-being has been derived from two general perspectives: the hedonic approach, which focuses on happiness and defines well-being in terms of pleasure attainment and pain avoidance; and the eudaimonic approach, which focuses on meaning and self-realization and defines well-being in terms of the degree to which a person is fully functioning." There is nothing inherent in any discussion of mindsets to suggest that the relationships in which we were interested would vary as a function of the type of well-being being analyzed. Accordingly, we expected that incrementalist beliefs would be positively related to both types of well-being.

## Mastery

We conceptualized mastery in terms of the definition of mastery offered by Ryff and colleagues [13] who describe mastery in terms of those high and low in mastery on their measure of mastery, which we used:

> "High scorer: has a sense of mastery and competence in managing the environment, controls complex array of external activities, makes effective use of surrounding opportunities, able to choose or create contexts suitable to personal needs and values. Low scorer, has difficulty managing everyday affairs, feels unable to change or improve surrounding context, is unaware of surrounding opportunities, lacks sense of control over external world" (p. 727).

People who are high in mastery feel competent and in control of their worlds. In numerous papers, Dweck and colleagues have asserted that a mastery orientation is responsible for relationship between positive outcomes and mindsets [11]. We believed that the mastery orientation described by Ryff and Keyes corresponded to the mastery orientation described by Dweck and colleagues, and our study examined this connection.

## Mastery, mindsets, and well-being

One of the cornerstones of Dweck and colleagues' model is that incrementalist mindsets are positively related to a mastery orientation, which is responsible for relationships between incrementalism and achievement [11]. An assumption of much of Dweck's work is that individuals with a more incrementalist mindset are more likely to view failure as a challenge and persist than individuals with a less incrementalist mindset. In turn, this greater persistence and sense of challenge

are assumed to result in improved performance. Moreover, research on mindsets is not limited to examining reactions to failure. For example, Martin [14] examined relationships between mindsets and growth (personal best) goals. Consistent with the broad outlines of previous research, he found that growth goals were positively related to incremental beliefs and were negatively related to entity beliefs.

The assumed sequence of mindset to performance is not discussed in detail in many studies. For example, Burnette et al. [4] emphasized the direct relationship between mindsets and self-regulatory processes and then relationships between self-regulatory processes and achievement (p. 657). The direct relationship between mindsets and performance is mentioned in passing, and the word "mediation" does not appear in this article. Similarly, Dweck and Yeager [15] discuss how mindsets are related to attributions, goals, and effort beliefs and how these three constructs are related to performance (p. 485), but they do not address whether such relationships are mediational or not. The word "mediation" does not appear in their article. Burnette et al. [5] focused solely on zero-order (direct) relationships between mindsets and psychological distress and did not address issues of mediation.

Nevertheless, our understanding of Dweck's basic models is that it is a mediational model in that relationships between mindsets and outcomes are presumed to operate through various intermediary constructs, one of which is mastery orientation. Moreover, there is some support for such a possibility. For example, Cypryańska and Nezlek [16] found that the satisfaction of the basic needs of autonomy and competence posited by Deci and Ryan's Self-Determination Theory [17] mediated relationships between incremental beliefs and well-being. Although not exactly the same as mastery orientation, the satisfaction of autonomy and competence needs, particularly competence, represent similar constructs, Along these same lines, Zhao et al. [18] found that grit mediated relationships between growth (incremental) mindsets and subjective well-being among high school students. Although grit is not mastery, grit refers to the motivation to persist in pursuing long term goals [19], which may be a precursor or correlate of mastery. Similarly, in a study of junior high school students, Zhao et al. [20] found that core self-evaluation mediated relationships between mindsets and meaning in life. Core self-evaluation is a construct that includes self-efficacy and locus of control, which are constructs that overlap conceptually with mastery.

For present purposes, we understand mediation in terms of three variables: an independent variable (or predictor) X, a dependent variable (or outcome) Y, and a mediator, M. Within the present context, mindset is the predictor, well-being is the outcome, and mastery orientation is the mediator. In the classic treatment of mediation offered by Baron and Kenny [21], there are two prerequisites to examine mediation: the predictor is related to the outcome (mindsets are related to well-being), and the predictor is related to the mediator (mindsets are related to mastery). The presence of mediation is determined by whether the outcome (well-being) is related to the predictor (mindsets) controlling for the mediator (mastery). There has been some discussion of the need for the first prerequisite [e.g., 22], but in the present case, we assumed that mindsets would be related to well-being. See Agler and De Boeck [23] for a discussion of when it is sensible to examine mediation in the absence of a direct effect between a predictor and an outcome.

Our general expectation was that mastery orientation would mediate relationships between mindsets and well-being. We expected this to occur for both hedonic and eudaimonic measures of well-being. We conducted these analyses with the knowledge that mediation, particularly within the context of a cross-sectional design with only one measurement occasion, does not provide a strong basis for drawing causal inferences. We discuss the issue of causal precedence in the discussion section.

## Method

### Participants and procedure

Participants were 447 students attending a university in Poland. Their average age was 27.2 years (SD = 7.93), and 86% were women. They participated in partial fulfillment of a course requirement. Data were collected between 12 October and 15 December 2021 using a secure website. Participants logged onto the study at this website, provided consent, and then completed the survey.



## Compliance with ethical standards

The study was conducted in accordance with the Declaration of Helsinki regarding the rights of research participants. Participants consented electronically by clicking on a link indicating their agreement to participate after being told that their names would not be associated with their answers and that they could terminate participation at any time without penalty. Consistent with these instructions, responses were de-identified before being provided to the researchers. The study was approved on 11 May 2021 by the Commission for Research Ethics, Faculty of Psychology in Warsaw, SWPS University, approval number: 44/2021.

## Measures

The measures we used were selected with the following criteria in mind. Most important, they needed to be psychometrically sound measures of the constructs in which we were interested, i.e., they needed to have demonstrated validity and reliability. They also needed to be appropriate for mass administration, i.e., relatively brief with easy to understand items and response formats.

## Mindsets

Mindsets were measured with a scale proposed by Dweck [11]. The scale has four items that measure incrementalist beliefs, e.g., "People can always substantially change the kind of person they are," and four items that measure entity beliefs, e.g., "Everyone is a certain kind of person, and there is not much that can be done to really change that." We used a Polish language version of this scale developed by Lachowicz-Tabaczek [24]. Responses were made using a six-point scale with endpoints labeled: 1 = definitely disagree and 6 = definitely agree. Consistent with previous practice, scale scores were defined as the mean response to all items with items that measured entity beliefs reverse-scored. This meant that higher scores represented holding more incremental vs. entity beliefs.

## Mastery orientation

Mastery was measured using the environmental mastery subscale of Ryff and Keyes's measure of well-being [13]. The scale has seven items. An example item from the environmental mastery subscale is "I am quite good at managing the many responsibilities of my daily life." We used a Polish language version of Ryff and Keyes's measure developed by Karaś and Cieciuch [25]. Responses were made using a seven-point scale with endpoints labeled: 1 = definitely disagree and 7 = definitely agree. Subscale scores were calculated following the scoring key provided by Ryff and Keyes [13].

## Well-being: Ryff and Keyes

Well-being was measured with the five other subscales of the measure developed by Ryff and Keyes [13]. These sub-scales were autonomy, personal growth, positive relations with others, purpose in life, and self-acceptance. The response scale was the same as that used for the environmental mastery subscale. Subscale scores were calculated following the scoring key provided by Ryff and Keyes [13].

## Well-being: positivity

Overall positivity was measured using the Positivity Scale [26], Polish adaptation by Łaguna et al. [27]. Responses were made using a five-point scale with endpoints labeled: 1 = strongly disagree and 5 = strongly agree. Scale scores were defined as the mean response to all items. Higher scores represented a more positive vs. less positive orientation.

## Well-being: self-esteem

Self-esteem was measured using Rosenberg's self-esteem scale [28], Polish adaptation by Łaguna et al. [29]. Responses were made using a five-point scale with endpoints labeled: 1 = strongly disagree and 5 = strongly agree. Scale scores were

defined as the mean response to all items, with negatively valent items reverse-scored. Higher scores represented higher vs. lower self-esteem.

### Well-being: loneliness

Loneliness was measured using the de Jong-Gierveld loneliness scale [30], Polish adaptation by Grygiel et al. [31]. Responses were made using a five-point scale with endpoints labeled: 1 = definitely not and 5 = definitely yes. To be consistent with the scoring of the other measures, this scale was scored so that higher scores on this scale represented greater well-being (i.e., the absence of loneliness or the presence of a social network).

### Well-being: optimism

Optimism was measured using the Revised Life Orientation Test [LOT-R; 32]. The research team, which had members fluent in English and Polish, created a Polish language version of the LOT-R for this study. Responses were made using a five-point scale with endpoints labeled: 1 = definitely not and 5 = definitely yes. The scale was scored using the key provided by Scheier et al. [32]. Higher scores indicated more vs. less optimism.

### Future time perspective

We measured participants' perceptions of their futures using a measure developed by Lang and Carstensen [33], Polish adaptation by Przepiorka et al. [34]. Responses were made using a seven-point scale with endpoints labeled: 1 = definitely not true and 7 = definitely true. The scale was scored following the guidelines proposed by Lang and Carstensen [33], i.e., as a single score with higher scores representing a more open vs. more closed future time perspective.

### Meaning in life

Meaning in life was measured using the Meaning in Life Questionnaire [35], Polish adaptation by Kossakowska et al. [36]. Responses were made using a seven-point scale with endpoints labeled: 1 = absolutely not true and 7 = absolutely true. Subscale scores for presence of meaning in life and search for meaning in life were calculated following the guidelines provided by Steger et al. [35]. Higher scores indicated more vs. less presence and search.

## Results

### Overview of analyses

Our primary interest was to determine if environmental mastery mediated relationships between mindsets and well-being. As discussed below, this was done using the PROCESS macro developed by Hayes [37]. We followed these mediational analyses with moderated mediation analyses to see if the mediational relationships we found varied as a function of participant sex. To provide a context for the primary analyses we examined the means, standard deviations, reliabilities, and correlations between our measures.

### Descriptive statistics

The means, standard deviations, reliabilities, and correlations between our measures are presented in Table 1. We estimated reliability using McDonald's omega. According to the guidelines suggested by Shrout [38], all measures had at least moderate reliability (.72 to.80), and most had substantial reliability (.81 and above).

### Relationships between mindsets and well-being and mastery

The correlations presented in Table 1 indicate that there was a positive relationship between well-being (broadly defined) and how incremental people's mindsets were. Scores on the measure of mindsets (higher scores represent a more

incrementalist mindset than lower scores) were positively related to all measures of well-being except for autonomy and search for meaning in life. Recall that loneliness was reverse-scored so higher scores represented being less lonely than lower scores.

We believed that understanding mediational relationships between mindsets and well-being (the focus of the next set of analyses) would be enhanced by understanding relationships between mastery and well-being. Considering such relationships is part of what Agler and De Boeck [23] refer to as the "An indirectness perspective," i.e., providing a fuller picture of relationships between predictors and outcomes. As can be seen from the correlations presented in Table 1, scores on the environmental mastery scale were positively (significantly) correlated with all measures of well-being, except for search for meaning in life.

### Environmental mastery as a mediator of relationships between incremental beliefs and well-being

The primary analyses of the study were a series of mediational analyses in which a measure of well-being was the outcome, the score on our mindsets measure was the predictor, and scores on the environmental mastery subscale of Ryff and Keyes's measure of well-being was the mediator. These analyses were done using the PROCESS macro, Model 4 [37]. In these analyses, we used bootstrapping (5,000 iterations) to obtain 99% confidence intervals (CI). The results of these analyses are summarized in Table 2. Effects for which the CI did not include 0 were interpreted as being significantly different from 0. To provide a basis to understand how total effects were partitioned into direct and indirect effects we present unstandardized coefficients.

The results of these analyses were clear. Environmental mastery mediated relationships between mindsets and all measures of well-being, with the exception of search for meaning in life. Putting aside debates about what constitutes full and partial mediation and how to quantify the strength of mediational relationships, when environmental mastery was included as a mediator, the direct effect for mindsets was rendered not significant (the CI included 0) in the analyses of positivity, self-esteem, loneliness, optimism, presence of meaning in life, purpose in life and self-acceptance. Moreover, the indirect effects for environmental mastery represented 50% or more of the total effects in the analyses of all measures except for search for meaning in life and autonomy. This, combined with the lack of significant direct effects for mindsets

**Table 1. Descriptive statistics, reliabilities, and correlations for measures.**

| Measure | M | SD | ω | Pos | RSE | Lone | LOT | FTP | MLQP | MLQS | E-Mast | Auto | Grow | P-Rel | Purp | S-Acc |
|---|---|---|---|---|---|---|---|---|---|---|---|---|---|---|---|---|
| Mindsets | 3.98 | .89 | .88 | .22 | .16 | .17 | .20 | .20 | .21 | .03 | .23 | .08 | .23 | .27 | .20 | .20 |
| Positivity | 3.81 | .71 | .89 | | .80 | .56 | .73 | .56 | .64 | .05 | .71 | .39 | .51 | .55 | .57 | .76 |
| RSE | 3.73 | .77 | .88 | | | .55 | .76 | .48 | .60 | −.01 | .68 | .46 | .52 | .47 | .53 | .80 |
| Loneliness | 3.43 | .54 | .80 | | | | .45 | .41 | .47 | −.02 | .63 | .31 | .36 | .71 | .43 | .62 |
| LOT | 3.61 | .84 | .86 | | | | | .50 | .56 | .05 | .62 | .36 | .47 | .43 | .48 | .70 |
| FTP | 4.77 | .92 | .80 | | | | | | .38 | .07 | .50 | .20 | .30 | .42 | .35 | .50 |
| MLQ Presence | 5.06 | .35 | .90 | | | | | | | .12 | .62 | .38 | .52 | .46 | .72 | .61 |
| MLQ Search | 5.35 | 1.14 | .84 | | | | | | | | −.06 | −.04 | .13 | .11 | .09 | −.08 |
| Environmental mastery | 4.68 | 1.09 | .83 | | | | | | | | | .48 | .54 | .62 | .63 | .81 |
| Autonomy | 4.80 | 1.02 | .80 | | | | | | | | | | .39 | .28 | .40 | .49 |
| Growth | 5.24 | .88 | .73 | | | | | | | | | | | .44 | .61 | .57 |
| Positive relationships | 5.38 | 1.00 | .82 | | | | | | | | | | | | .44 | .61 |
| Purpose | 5.18 | .91 | .72 | | | | | | | | | | | | | .58 |
| Self acceptance | 4.82 | .14 | .86 | | | | | | | | | | | | | |

Note: For $n = 447$: $|r| > .13$, $p < .01$; $|r| > .10$, $p < .05$.



**Table 2. Summary of mediation analyses: Environmental mastery as a mediator of relationships between mindsets and well-being.**

| Measure | total effect | direct effect | Indirect effect | % indirect |
|---|---|---|---|---|
| Positivity | .175 (.080/.270) | .048 (−.022/.119) | .127 (.030/.204) | 72.6% |
| RSE | .138 (.033/.242) | .003 (−.077/.083) | .135 (.058/.223) | 97.8% |
| Loneliness | .102 (.029/.175) | .015 (−.044/.074) | .087 (.037/.143) | 85.3% |
| LOT | .193 (.079/.306) | .061 (−.032/.155) | .131 (.053/.211) | 67.9% |
| FTP | .208 (.084/.332) | .094 (−.018/.207) | .114 (.045/.191) | 54.8% |
| MLQ Presence | .315 (.133/.497) | .104 (−.046/.254) | .211 (.088/.339) | 67.0% |
| MLQ Search | .042 (−.115/.200) | .064 (−.097/.226) | −.022 (−.066/.018) | |
| Autonomy | .087 (−.053/.227) | −.042 (−.168/.084) | .129 (.055/.211) | |
| Growth | .222 (.104/.339) | .106 (.002/.210) | .116 (.048/.190) | 52.3% |
| Positive relationships | .303 (.171/.436) | .151 (.042/.261) | .152 (.061/.249) | 50.2% |
| Purpose | .203 (.080/.326) | .059 (−.042/.159) | .144 (.059/.233) | 70.9% |
| Self-acceptance | .259 (.106/.413) | .024 (−.071/.119) | .236 (.095/.368) | 91.1% |

when environmental mastery was included as a mediator, provides strong support for the central hypothesis of the study that relationships between incremental mindsets and well-being go through environmental mastery.

Finally, although there were no theoretical reasons or prior research suggesting that the relationships that were the focus of this study should differ between women and men, the sample did not have a balanced number of men and women. In light of this, we conducted moderated mediation analyses which were the same analyses as described above with sex added as moderating variables. This was Model 5 within the nomenclature of PROCESS. These analyses did not find any significant moderation effects.

## Discussion

The results supported our hypothesis that a sense of mastery mediates relationships between incremental mindsets and well-being. We found that environmental mastery mediated relationships between incremental beliefs and well-being for all measures of well-being except for search for meaning in life. Moreover, this mediation was strong. For most measures of well-being, the direct effect between mindsets and well-being was not significant after environmental mastery was included as a mediator, something that was initially referred to as "full mediation" [21] but might be better understood as "indirect only mediation" [39]. Regardless, these results support our logic that the sense of mastery that has been discussed as one of the important reasons incrementalist beliefs are associated with success also applies to relationships between incrementalist beliefs and well-being.

The present results constitute an important, explicit confirmation of a critical assumption of mindset theory. Previous research has examined components of the present mediational sequence, but we are unaware of research that has examined explicitly how mastery mediates relationships between mindsets and well-being. Moreover, the formal mediational analyses we conducted provided a basis to estimate indirect effects and the extent to which these indirect effects accounted for total effects.

### Causal precedence

Finding a mediational relationship, even a strong one, within the context of a cross-sectional, single-occasion design cannot serve as a basis for claims about causal relationships among the constructs being measured. Nevertheless, the fact that M mediates a relationship between X and Y can be interpreted as being consistent with, or supportive of, the existence of a causal sequence from X to M to Y. Such support can be meaningful when a mediational model is theory-driven, and when the components of the model represent relationships that have been supported by past research.

Although the results of experimental studies in which mindsets have been manipulated have been weak (at best) in terms of improving academic performance [e.g., 3], some research suggests that manipulating mindsets can lead to improvements in well-being. For example, Howell et al. [8; Study 4] found that temporarily increasing incrementalist beliefs led to greater endorsement of therapeutic lifestyle changes. Schleider and Weisz [40] found that a single 30-min. mindset growth intervention led to improvements in well-being measured nine months after the intervention. These studies suggest that changes in mindsets can lead to changes in well-being, but this is far from supporting evidence for the full theoretical model that served as the basis for the mediational analyses conducted in this study.

Nevertheless, we think the present results provide credible support for the existence of a causal sequence from incremental beliefs through mastery to well-being. The direct relationships we found between incremental beliefs and well-being and between incremental beliefs and mastery replicate previous research, and they are consistent with the general outlines of research on mindsets. Nonetheless, the existence of mediation can be used to support, but not prove causality. Causality is best examined using experimental methods or studies of relationships across time when it is difficult to manipulate people's levels of constructs.

### Other approaches to mediation involving mindsets and well-being

The mediational model we tested, that mastery would mediate relationships between mindsets and well-being, was based directly on the original theorizing of Dweck and colleagues and was based on subsequent research. Nevertheless, other researchers have examined other mediational relationships involving mindsets and well-being, and for the sake of thoroughness, we briefly discuss them and their relevance to the current results.

For example, Gál et al. [41] found that self-esteem mediated relationships between entity mindsets and negative emotions, although their study examined retrospective reports of a specific academic failure, and they did not measure growth (incrementalist) mindsets. Lam and Zhou [42] found that life satisfaction and perceived distress mediated relationships between growth mindsets and perseverance. Lam and Zhou did not measure entity mindsets.

### Limitations and future directions

The present study examined relationships among mindsets, well-being, and mastery among a sample of Polish university students. Although we have no reason to believe that there is anything about Poles that distinguish them from members of other Western cultures in ways that are relevant to the issues at hand, there may be. There is also the issue of how mindsets function in non-Western cultures. Some research, e.g., Sun et al. [43], suggests that mindsets may function differently in Asian societies than they do in Western societies.

In contrast, other research, e.g., Huang et al. [6] and Zhao et al. [18], suggests that the relationships between mindsets and outcomes in Western and Asian cultures are similar. Both of these studies found positive relationships between growth mindsets (incrementalism) and well-being. Moreover, Zhao et al. [18,20] in studies of Chinese students found mediational relationships that were consistent conceptually with the present results.

The present study focused on the mediating role of mastery orientation, and mastery is only one of numerous possibilities. Dweck and colleagues have discussed various mediators of relationships between mindsets and performance/achievement such as attributions, types of goals, and effort beliefs [15]. Although they did not conduct mediation per se, Burnette et al. [4] described a mediation-like sequence of relationships involving regulatory processes. Moreover, it is not clear how these different potential mediators can/should be treated together. For example, Robins and Pals [7] presented a serial model in which various constructs between mindsets and self-esteem were ordered in a certain sequence, with goal orientation leading to mastery. We found that mastery fully mediated relationships between mindsets and most of our measures of well-being, but this does not preclude the possibility that other mediators such as grit or self-efficacy may play important roles. Disentangling such complexities will require research explicitly designed to do so.



## Conclusions

As expected, the present study found that a mastery orientation mediated relationships mindsets and well-being. This mediation occurred for various measures of well-being, both hedonic and eudaimonic. Moreover, by any standard, this mediation was strong. Although such mediation cannot be used to draw firm conclusions about causality, the results we found are consistent with a causal sequence from incrementalism to mastery to well-being.

When considering the present results, it must be kept in mind that the research suggests that changing people's mindsets is not easy [e.g., 3]. Although incrementalism in mindsets may be positively related to well-being, if one wants to increase or enhance people's well-being, it might be worthwhile to reconsider how effective or efficient interventions to change mindsets are. Our results suggest that relationships between mindsets and well-being "go through" environmental mastery. If this is the case, perhaps researchers and practitioners should focus on enhancing or improving people's environmental mastery rather than changing mindsets because environmental mastery may be a more proximal cause of well-being than mindsets are.

Consistent with this suggestion, there is a growing body of research indicating that environmental mastery can be improved, apparently more robustly than the extent to which mindsets can be improved. For example, in a meta-analysis of 77 studies, van Dierendonck and Lam [44] found that environmental mastery improved as a result of various interventions. Van Dierendock and Lam did not examine mediational relationships – they treated environmental mastery as a measure of well-being per se, which is understandable and in keeping with Ryff's original intent. Nevertheless, it is possible that Ryff's constructs of well-being can be arranged in some type of causal order with environmental mastery having a stronger causal precedence than the other measures of well-being.

It may turn out that changing individuals' sense of environmental mastery leads to changes in their mindsets. If so, the extent to which such increases in incrementalist beliefs matter in terms of well-being and performance above and beyond changes in mastery would need to be established. At this point, such possibilities are speculative and examining them will require studies that examine changes over time in mindsets, well-being, and environmental mastery.

## Author contributions

**Conceptualization:** John B. Nezlek, Marzena Cypryanska, Joanna Gutral.

**Data curation:** John B. Nezlek, Marzena Cypryanska, Joanna Gutral.

**Formal analysis:** John B. Nezlek.

**Project administration:** John B. Nezlek.

**Resources:** Marzena Cypryanska.

**Writing – original draft:** John B. Nezlek, Marzena Cypryanska, Joanna Gutral.

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
