## [Decision Letter · Decision Letter 0]

Dear Dr. Nezlek,

Thank you for submitting your manuscript to PLOS ONE. After careful consideration, we feel that it has merit but does not fully meet PLOS ONE’s publication criteria as it currently stands. Therefore, we invite you to submit a revised version of the manuscript that addresses the points raised during the review process.

Indeed, the manuscript requires significant revision as it demonstrates gaps in theoretical grounding, lacks clarity in defining key terms, and presents insufficient methodological details, making it difficult to assess the validity of the findings. Moreover, critical aspects such as the psychometric properties of measures and corrections for statistical errors are either omitted or inadequately addressed. These issues, besides an unclear articulation of the study's contribution to existing research, limit the manuscript's readiness for publication. Therefore, I invite you to revise your paper very cautiously.

We look forward to receiving your revised manuscript.

Kind regards,

Ali B. Mahmoud, Ph.D.

Academic Editor

PLOS ONE

Journal Requirements:

Reviewers' comments:

Reviewer's Responses to Questions

**Comments to the Author**

1. Is the manuscript technically sound, and do the data support the conclusions?

Reviewer #1: Yes

2. Has the statistical analysis been performed appropriately and rigorously?

Reviewer #1: I Don't Know

3. Have the authors made all data underlying the findings in their manuscript fully available?

Reviewer #1: Yes

4. Is the manuscript presented in an intelligible fashion and written in standard English?

Reviewer #1: Yes

Reviewer #1: Thank you for the opportunity to review this manuscript. While the topic is of interest and importance, I regret to say that I cannot support the publication of the paper in its current form. Below, I provide detailed feedback outlining significant concerns that must be addressed for the manuscript to be considered for publication elsewhere.

The theoretical foundation of the manuscript is weak. It is unclear whether the authors conducted a thorough literature review. For example, the relationship between mindset and achievement is well-documented as being mediated by other factors, such as motivation, grit, and self-efficacy (e.g., Zhao et al., 2024). However, the manuscript implies that these mediational relationships have rarely been studied, which is inaccurate. A more comprehensive review of the literature and clearer articulation of the study's unique contribution are necessary.

Several key terms, such as "treatment value" and "entity orientation" (p. 4), are introduced without sufficient explanation or context. Providing definitions and background for these terms is crucial for clarity. Additionally, the inclusion of correlation coefficients from Burnette et al. on the same page seems unnecessary unless their relevance to the current study is explicitly explained. Furthermore, the phrase "a series of studies" lacks specificity. If the authors are referring to multiple studies, these should be listed or described. Simply citing a single study is inadequate to substantiate the claim.

The manuscript lacks critical details about the study methods, which significantly undermines its credibility. There is no dedicated data analysis section, leaving readers unable to evaluate how the results were derived. The authors should provide structured and transparent information about their study design, data collection, and analysis plan.

Additionally, the psychometric properties of the measures used in the study are not addressed. Can the validity and reliability of these measures be assumed? If not, how was this assessed? Without such information, the results are difficult to evaluate. Another major concern is the lack of attention to potential Type I error inflation due to multiple tests. Have any corrections (e.g., Bonferroni adjustments) been applied? If not, this is a significant limitation that compromises the integrity of the findings.

In its current form, the manuscript lacks the rigor and clarity required for publication. Addressing the above issues would require substantial revisions, including a more thorough literature review, clearer definitions and context, and detailed reporting of study methods and analyses. I appreciate the authors' efforts but must recommend rejection.

Thank you for considering my feedback.

**Do you want your identity to be public for this peer review?** For information about this choice, including consent withdrawal, please see our Privacy Policy

Reviewer #1: No

---

## [Author Response · Author response to Decision Letter 1]

12 Feb 2025

Responses to reviewers uploaed as a separate file/document.

---

## [Decision Letter · Decision Letter 1]

Dear Dr. Nezlek,

Thank you for submitting your manuscript to PLOS ONE. After careful consideration, we feel that it has merit but does not fully meet PLOS ONE’s publication criteria as it currently stands. Therefore, we invite you to submit a revised version of the manuscript that addresses the points raised during the review process.

We look forward to receiving your revised manuscript.

Kind regards,

Ali B. Mahmoud, Ph.D.

Academic Editor

PLOS ONE

Reviewers' comments:

Reviewer's Responses to Questions

**Comments to the Author**

Reviewer #1: All comments have been addressed

2. Is the manuscript technically sound, and do the data support the conclusions?

Reviewer #1: Yes

3. Has the statistical analysis been performed appropriately and rigorously?

Reviewer #1: Yes

4. Have the authors made all data underlying the findings in their manuscript fully available?

Reviewer #1: Yes

5. Is the manuscript presented in an intelligible fashion and written in standard English?

Reviewer #1: Yes

**Reviewer #1: ** Thank you for your thoughtful and detailed revision. I appreciate the work you’ve done to improve the manuscript, and I can see that it's moving in a good direction. Your responses to the initial feedback were helpful, and the revised version reflects meaningful progress.

At this point, I recommend a major revision to further strengthen the clarity and rigor of the analysis and interpretation.

One area that still needs attention is the construct validity of the outcome measures. My earlier comment on psychometric properties may not have been clear, and I apologize for that. In addition to reporting reliability (e.g., internal consistency), it’s important to consider whether the measures are valid representations of the constructs they intend to capture. If item-level data are available, I encourage you to conduct and report basic checks—such as confirmatory factor analyses—to support the use of the measures. If that’s not possible due to data limitations, it would be helpful to state this and acknowledge it as a limitation.

Also, I noticed the mention of the Sobel test in relation to controlling Type I error. Just to clarify, the Sobel test is not intended for that purpose, and its assumptions may not hold in many mediation contexts. Your use of bootstrapping is a more appropriate approach and aligns well with current best practices. However, since you're conducting multiple mediation models, I suggest considering a more conservative approach to confidence intervals—such as using a 99% CI instead of 95%, following a Bonferroni-type adjustment. I marked my recommendation as a major revision since this change in confidence level may impact the significance of your findings.

I hope this helps clarify the remaining points.

**Do you want your identity to be public for this peer review?** For information about this choice, including consent withdrawal, please see our Privacy Policy

Reviewer #1: No

---

## [Decision Letter · Decision Letter 2]

Environmental mastery mediates relationships

between mindsets and well-being

PONE-D-24-45317R2

Dear Dr. Nezlek,

We’re pleased to inform you that your manuscript has been judged scientifically suitable for publication and will be formally accepted for publication once it meets all outstanding technical requirements.

Kind regards,

Ali B. Mahmoud, Ph.D.

Academic Editor

PLOS ONE

Additional Editor Comments (optional):

Reviewers' comments:

Reviewer's Responses to Questions

**Comments to the Author**

Reviewer #1: All comments have been addressed

2. Is the manuscript technically sound, and do the data support the conclusions?

Reviewer #1: Yes

3. Has the statistical analysis been performed appropriately and rigorously?

Reviewer #1: Yes

4. Have the authors made all data underlying the findings in their manuscript fully available?

Reviewer #1: Yes

5. Is the manuscript presented in an intelligible fashion and written in standard English?

Reviewer #1: Yes

Reviewer #1: (No Response)

**Do you want your identity to be public for this peer review?** For information about this choice, including consent withdrawal, please see our Privacy Policy

Reviewer #1: No

---

## [Editor Report · Acceptance letter]

PONE-D-24-45317R2

PLOS ONE

Dear Dr. Nezlek,

I'm pleased to inform you that your manuscript has been deemed suitable for publication in PLOS ONE. Congratulations! Your manuscript is now being handed over to our production team.

Kind regards,

on behalf of

Dr. Ali B. Mahmoud

Academic Editor

PLOS ONE